# The Initial Impacts of the COVID-19 Pandemic on Regional Economies in Indonesia: Structural Changes and Regional Income Inequality

Takahiro Akita [1,*] and Armida Salsiah Alisjahbana [2]

[1]  IUJ Research Institute, International University of Japan, Minami-Uonuma, Niigata 949-7277, Japan
[2]  Faculty of Economics and Business, Universitas Padjadjaran, Bandung 45363, West Java, Indonesia; armida.alisjahbana@unpad.ac.id
*   Correspondence: akita@iuj.ac.jp; Tel.: +81-90-2728-0272

**Abstract:** The COVID-19 pandemic has exerted an enormous impact on the Indonesian economy. In 2020, the country's economy contracted by 2.7%. However, the impact has been spatially heterogeneous. Based on provincial GDP across industrial sectors, this study examines how structural changes caused by the pandemic have affected the determinants of inter-provincial inequality in Indonesia by conducting a bi-dimensional inequality decomposition analysis. According to the bi-dimensional decomposition analysis, after the outbreak of COVID-19, the tourism sector reduced its contribution to inter-provincial inequality. On the other hand, the IC and financial services sectors were not affected by the pandemic and raised their contributions. When Indonesia recovers from the pandemic, it is likely that the tourism sector will regain its position as an important determinant of inter-provincial inequality. However, the most important sectors in determining inter-provincial inequality will be the IC, financial, and business services sectors, particularly in the Java–Bali region. With the rapid advancement of IC, financial, and e-business technologies, the roles of these high-inequality sectors are likely to increase unless policies that could facilitate spatial dispersion of these services and activities are implemented.

**Keywords:** Indonesia; COVID-19 pandemic; structural changes; inter-provincial income inequality; bi-dimensional inequality decomposition analysis

## 1. Introduction

Since the outbreak of the novel coronavirus disease (COVID-19) in early 2020, the number of confirmed COVID-19 cases has increased exponentially in Indonesia. As of August 2021, 4.1 million people have been infected, meaning that at least 15 out of 1000 people have been infected. The COVID-19 pandemic hit the Indonesian economy severely. In 2020, real GDP decreased by 2.7% (Figure 1). Since the annual average growth rate was 5.3% between 2010 and 2019, the impact of the COVID-19 pandemic has been enormous. The COVID-19 pandemic had, however, differential impacts on regional economies (Figure 2). While tourist destination provinces such as Bali and Riau Islands experienced a large decrease in per capita GDP, the impact appears to have been relatively small in such eastern provinces as Central Sulawesi, North Maluku and Papua (see Figure 3 for a map of Indonesia).

How has the COVID-19 pandemic affected regional economies? How have structural changes caused by the COVID-19 pandemic affected the determinants of regional income inequality? This study addresses these questions using provincial GDP for industrial sectors in Indonesia. It uses a bi-dimensional inequality decomposition method to explore the determinants of inter-provincial inequality in per capita GDP before and after the outbreak of COVID-19. The bi-dimensional inequality decomposition method employs a squared population-weighted coefficient of variation (squared *WCV*) as a measure of inequality and decomposes inter-provincial inequality in per capita GDP along two dimensions: by region

and by GDP components (The population-weighted coefficient of variation, introduced by Williamson [2], has been used by many researchers to measure regional income inequality. See, for example, Mathur [3], Tabuchi [4], Mutlu [5], Akita and Lukman [6], Fujita and Hu [7], and Hill and Vidyattama [8]). The squared *WCV* satisfies several desirable properties as a measure of inequality, such as anonymity, income homogeneity, population homogeneity and the Pigou–Dalton transfer principle (Anand, [9]; Fields [10]). Since the squared *WCV* is a member of the generalized entropy class of inequality measures, inter-provincial inequality in per capita GDP, as measured by the squared *WCV*, can be decomposed additively by region; that is, decomposed into within-region and between-region inequality components (Shorrocks [11]) (Here, provinces are classified into mutually exclusive and collectively exhaustive regions.). Furthermore, using the squared *WCV*, inter-provincial inequality can be decomposed by GDP components; that is, expressed as the sum of contributions from GDP components (Shorrocks, [12]) (Here, total GDP consists of several GDP components (GDP from several industrial sectors).). Using the squared *WCV*, the bi-dimensional inequality decomposition method combines these two decomposition properties. It can thus analyze the contribution of each GDP component to overall inter-provincial inequality in GDP per capita through within-region and between-region inequalities in a coherent framework.

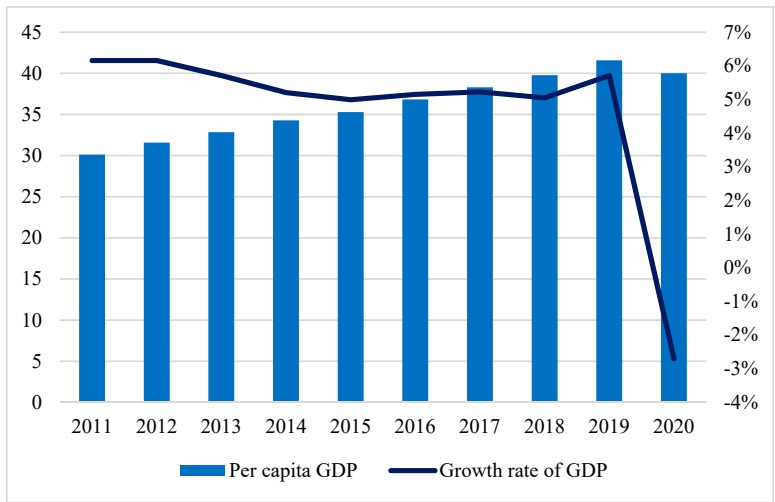

**Figure 1.** Growth Rate of GDP and per capita GDP (at constant 2010 prices), 2010–2020. Note: Per capita GDP is in million Rupiah. Source: Authors' calculation based on Central Bureau of Statistics [1].

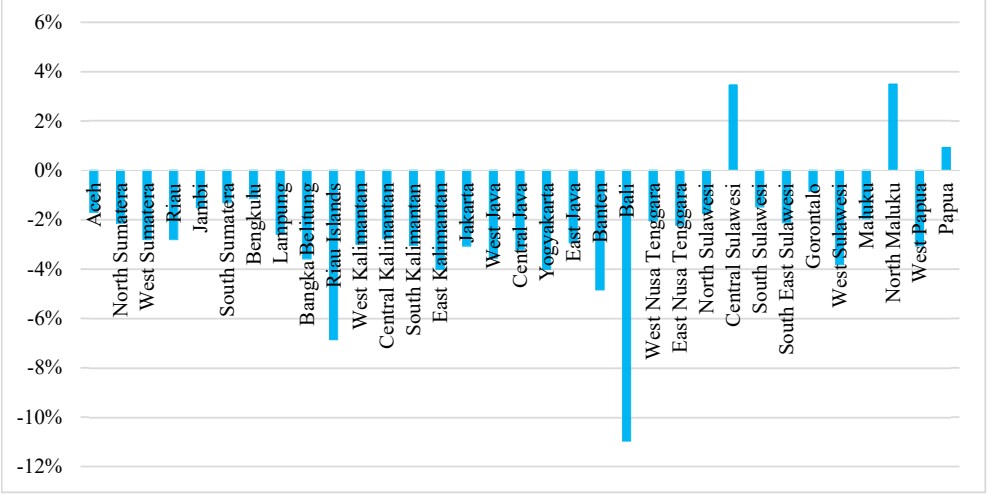

**Figure 2.** Growth Rate of per capita GDP by province, 2019–2020. Source: Authors' calculation based on Central Bureau of Statistics [1].

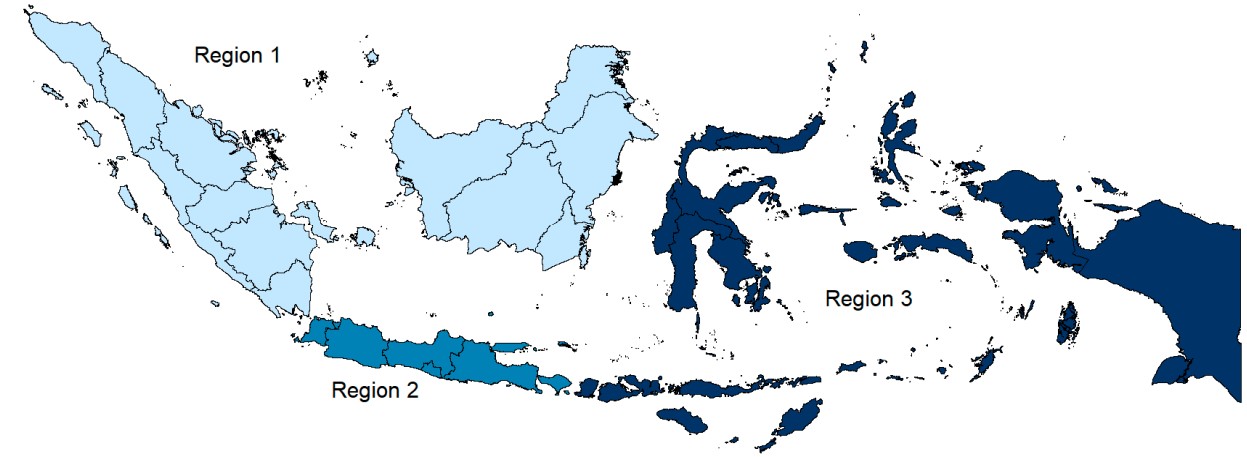

| Region | Province Code | Province | Region | Province Code | Province |
|---|---|---|---|---|---|
| Region 1 | 11 | Aceh | Region 2 | 31 | Jakarta |
| | 12 | North Sumatera | | 32 | West Java |
| | 13 | West Sumatera | | 33 | Central Java |
| | 14 | Riau | | 34 | Yogyakarta |
| | 15 | Jambi | | 35 | East Java |
| | 16 | South Sumatera | | 36 | Banten |
| | 17 | Bengkulu | | 51 | Bali |
| | 18 | Lampung | Region 3 | 52 | West Nusa Tenggara |
| | 19 | Bangka Belitung | | 53 | East Nusa Tenggara |
| | 21 | Riau Islands | | 71 | North Sulawesi |
| | 61 | West Kalimantan | | 72 | Central Sulawesi |
| | 62 | Central Kalimantan | | 73 | South Sulawesi |
| | 63 | South Kalimantan | | 74 | S.E. Sulawesi |
| | 64 | East Kalimantan | | 75 | Gorontalo |
| | | | | 76 | West Sulawesi |
| | | | | 81 | Maluku |
| | | | | 82 | North Maluku |
| | | | | 91 | West Papua |
| | | | | 94 | Papua |

**Figure 3.** Map of Indonesia. Note: Classification of provinces.

The next section provides a literature review pertaining to our study. Section 3 presents the data and the methods used in this study. Section 4 discusses the results, while Section 5 provides concluding remarks.

## 2. Literature Review

To the best of our knowledge, only a few studies have investigated the impact of the COVID-19 pandemic on the distribution of income in Indonesia. (Following the completion of this paper, several additional articles addressing the impacts of the COVID-19 on income inequality have been published. They include Brata et al. [13], Suryahadi, et al. [14], and Novianti and Panjaitan [15], but they differ from our study in both methodology and data.) Suryahadi, et al. [16] estimated the impact of the pandemic on poverty in Indonesia by conducting a simulation analysis based on the past pattern of economic shocks. They found that under the worst-case scenario of economic growth, in which the economy contracts by 3.5%, the poverty headcount ratio increases from 9.2% in 2019 to 16.6% by the end of 2020. This means that 19.7 million people become poor, bringing the country back to 2004, when the poverty headcount ratio was 16.7%.

Gibson and Olivia [17] also investigated the impact of the COVID-19 pandemic on poverty in Indonesia. Unlike Suryahadi, et al. [16], they estimated the impact at the provincial level using mobility data from Google. They found that the impact varied substantially across provinces. Provinces with lower initial poverty headcount ratios tended to have a larger increase in the headcount ratio. For example, in Bali, one of the richest provinces, the poverty headcount ratio increased by 13 percentage points, while in the poorest provinces of Papua and East Nusa Tenggara, it increased by 3 percentage points. They thus argued that the social assistance program needed to be expanded in places where people had not widely relied on it previously.

On the other hand, numerous studies have been conducted to analyze regional income inequality in Indonesia using provincial GDP because a large income disparity has persisted between provinces. (In 2022, Jakarta had the largest per capita GDP at 298 million Rupiah, which was 14 times the smallest in East Nusa Tenggara.) These studies include Esmara [18], Akita and Lukman [6], Garcia and Soelistianingsih [19], Hill, et al. [20], Vidyattama [21], Hill and Vidyattama [8], and Alisjahbana and Akita [22].

Akita and Lukman [6] used provincial GDP by industrial sector to explore the determinants of inequality in per capita GDP from 1975 to 1992. They conducted an inequality decomposition analysis by industrial sector using the *WCV*. Hill and Vidyattama [8] used an updated dataset of provincial GDP to analyze inequality in per capita GDP from 1975 to 2010. On the other hand, Garcia and Soelistianingsih [19] investigated $\beta$-convergence using provincial GDP from 1975 to 1993. Hill, et al. [20] also examined $\beta$-convergence using an updated provincial GDP data set from 1975 to 2002. Vidyattama [21] employed a spatial econometric approach to investigate the impact of the neighborhood effect on the speed of $\beta$-convergence using provincial- and district-level GDP from 1999 to 2008.

Alisjahbana and Akita [22] used provincial GDP by industrial sector from 2005 to 2013 to examine how economic tertiarization and concurrent output deindustrialization have affected the determinants of inter-provincial inequality in per capita GDP by conducting a bi-dimensional inequality decomposition analysis. Our study is similar to this study in terms of method, involving the assessment of regional sustainability from two dimensions: economic and spatial development (see Shmelev and Shmeleva [23]). But, it uses provincial GDP by industrial sector from 2010 to 2020 and analyzes the initial impacts of the COVID-19 pandemic on inter-provincial inequality in per capita GDP.

Our study also differs from that of Alisjahbana and Akita [22] in that it uses a 52-sector classification, while Alisjahbana and Akita [22] used 33-sector classifications (see Table 1 for the sector classifications). Thus, our study could analyze the impact of structural changes on inter-provincial inequality in greater detail. With 33-sector classifications, we are not able to analyze structural changes within the manufacturing sector. In the 33-sector classification, the manufacturing sector consists of two subsectors: oil and gas manufacturing and non-oil and non-gas manufacturing, while it contains 16 subsectors in the 52-classification (see Table 1). The manufacturing sector has served as the engine of growth and played an important role in regional economies. With 16 subsectors, therefore, our study provides much better insights into the role of the manufacturing sector in inter-provincial income inequality.

**Table 1.** Sector classifications.

| | 9 Main Sectors | Manufacturing and Services | | 52 Sectors |
|---|---|---|---|---|
| 1 | Agriculture | | 1 | Food crops |
| | | | 2 | Horticultural crops |
| | | | 3 | Plantation crops |
| | | | 4 | Livestock |
| | | | 5 | Agriculture services and hunting |
| | | | 6 | Forestry and logging |
| | | | 7 | Fishery |

**Table 1.** *Cont.*

| 9 Main Sectors | | Manufacturing and Services | | 52 Sectors | |
|---|---|---|---|---|---|
| 2 | Mining | | | 8 | Crude petroleum, natural gas, and geothermal |
| | | | | 9 | Coal and lignite mining |
| | | | | 10 | Iron ore mining |
| | | | | 11 | Other mining and quarrying |
| 3 | Manufacturing | m1 | Coal and refined petroleum products | 12 | Coal and refined petroleum products |
| | | m2 | Food, tobacco and beverages | 13 | Food products and beverages |
| | | | | 14 | Tobacco products |
| | | m3 | Textiles, wearing apparel and leather products | 15 | Textiles and wearing apparel |
| | | | | 16 | Leather and related products and footwear |
| | | m4 | Wood products, furniture and paper products | 17 | Wood products and cork |
| | | | | 18 | Paper products and printing |
| | | | | 19 | Furniture |
| | | m5 | Chemical, rubber and other non-metallic mineral products | 20 | Chemical products |
| | | | | 21 | Rubber and plastics products |
| | | | | 22 | Other non-metallic mineral products |
| | | m6 | Basic metals and fabricated metal products | 23 | Basic metals |
| | | | | 24 | Fabricated metal and optical products and computers |
| | | m7 | Machinery and equipment | 25 | Machinery and equipment |
| | | m8 | Transport equipment | 26 | Transport equipment |
| | | m9 | Other manufacturing | 27 | Other manufacturing products |
| 4 | Electricity/gas/water | | | 28 | Electricity |
| | | | | 29 | Gas |
| | | | | 30 | Water |
| 5 | Construction | | | 31 | Construction |
| 6 | Trade/hotel/restaurant | t1 | Wholesale and retail trade | 32 | Wholesale and retail trade |
| | | | | 33 | Other wholesale and retail trade |
| | | t2 | Hotels and restaurants | 34 | Hotels |
| | | | | 35 | Restaurants |
| 7 | Transportation/communication | t3 | Railway transportation | 36 | Railway transportation |
| | | t4 | Land transportation | 37 | Land transportation |
| | | t5 | Sea transportation | 38 | Sea transportation |
| | | t6 | River and lake transportation | 39 | River and lake transportation |
| | | t7 | Air transportation | 40 | Air transportation |
| | | t8 | Support services for transportation | 41 | Support services for transportation |
| | | t9 | Information and communication | 42 | Information and communication |
| 8 | Financial and business services | s1 | Financial intermediary services | 43 | Financial intermediary services |
| | | s2 | Insurance and pension fund | 44 | Insurance and pension fund |
| | | s3 | Other financial services | 45 | Other financial services |
| | | | | 46 | Financial supporting services |
| | | s4 | Real estate | 47 | Real estate |
| | | s5 | Business services | 48 | Business services |
| 9 | Other services | s6 | Public administration | 49 | Public administration and defense |
| | | s7 | Education services | 50 | Education |
| | | s8 | Health services | 51 | Health and social work |
| | | s9 | Other services | 52 | Other services |

### 3. Data and Methods

*3.1. Data*

This study used provincial GDP for 52 industrial sectors for the period from 2010 to 2020, compiled by the Indonesian Central Bureau of Statistics (CBS [1]). The data set includes GDP at constant 2010 prices for 33 provinces. In a bi-dimensional inequality decomposition analysis, these 33 provinces are divided into three mutually exclusive and collectively exhaustive regions: region 1 (Sumatra and Kalimantan provinces); region 2 (Java provinces and Bali); and region 3 (West and East Nusa Tenggara, Sulawesi provinces, Maluku, North Maluku, West Papua, and Papua) (see Figure 3).

A bi-dimensional inequality decomposition analysis was performed first using 9 main sectors, which are created by aggregating 52 sectors. These main sectors are (1) agriculture; (2) mining; (3) manufacturing; (4) electricity, gas, and water; (5) construction; (6) trade, hotel, and restaurant; (7) transportation and communication; (8) financial and business services; and (9) government and other services (see the first column in Table 1). Since the manufacturing and services sectors play an important role in determining inter-provincial inequality in per capita GDP, in the second step, we conducted decomposition analyses for (1) manufacturing subsectors; (2) trade, transportation, and IC (information and communication) subsectors; and (3) finance, business, and government services subsectors (see the second column in Table 1).

*3.2. Method: Bi-Dimensional Inequality Decomposition Method*

To analyze the effects of changes in industrial and spatial structures on inter-provincial inequality in per capita GDP, we conducted a bi-dimensional inequality decomposition analysis using the squared population-weighted coefficient of variation (squared *WCV*). The method we employed enabled us to comprehensively assess the effects of changes in both industrial and spatial structure on regional income inequality. Consequently, it offers a robust framework for examining the effects of COVID-19 on inter-provincial income inequality.

Suppose that a country consists of $m$ regions, and region $i$ is composed of $n_i$ provinces. Let $y_{ij}$, $p_{ij}$, $y$, and $p$ be, respectively, per capita GDP and population of province $j$ in region $i$, and per capita GDP and total population of the country. Then, inter-provincial inequality in per capita GDP can be measured using the following squared *WCV*.

$$WCV^2 = \frac{1}{y^2} \sum_{i=1}^{m} \sum_{j=1}^{n_i} \frac{p_{ij}}{p} \left(y_{ij} - y\right)^2, \tag{1}$$

where $y = \sum_{i=1}^{m} \sum_{j=1}^{n_i} \frac{p_{ij}}{p} y_{ij}$.

Let $y_i$, $p_i$, and $WCV_i^2 = \frac{1}{y_i^2} \sum_{j=1}^{n_i} \frac{p_{ij}}{p_i} \left(y_{ij} - y_i\right)^2$ be, respectively, per capita GDP, population, and the squared population-weighted coefficient of variation for region $i$. Then, the squared *WCV* can be decomposed into within- and between-region inequality components as follows (Shorrocks [11]).

$$WCV^2 = WCV_W + WCV_B. \tag{2}$$

In Equation (2), $WCV_W = \sum_{i=1}^{m} \left(\frac{p_i}{p}\right) \left(\frac{y_i}{y}\right)^2 WCV_i^2$ is the within-region inequality component, while $WCV_B = \frac{1}{y^2} \sum_{i=1}^{m} \frac{p_i}{p} \left(y_i - y\right)^2$ is the between-region inequality component. It should be noted that $WCV_W$ is not a weighted average of $WCV_i^2$, since the weights, $\left(\frac{p_i}{p}\right) \left(\frac{y_i}{y}\right)^2$, do not sum to unity.

Suppose next that total provincial GDP is composed of $K$ GDP components (GDP from $K$ industrial sectors); that is, $y_{ij} = \sum_{k=1}^{K} y_{ijk}$, where $y_{ijk}$ is per capita GDP from component $k$

of province $j$ in region $i$. Since squared $WCV$ can also be decomposed by GDP components, region $i$'s within-region inequality can be expressed as follows (Shorrocks [12]).

$$WCV_i^2 = \sum_{k=1}^{K} w_{ik} WCOV_{ik}.$$ (3)

In Equation (3), $w_{ik}$ is the GDP share of component $k$ in region $i$, while $WCOV_{ik} = \frac{1}{y_i y_{ik}} \sum_{j=1}^{n_i} \frac{p_{ij}}{p_i} (y_{ij} - y_i)(y_{ijk} - y_{ik})$ is the population-weighted coefficient of covariation between total per capita GDP and per capita GDP from component $k$ in region $i$, where, $y_{ik}$ is per capita GDP from component $k$ in region $i$.

Similarly, the between-region inequality can be decomposed by GDP components as follows:

$$WCV_B = \sum_{k=1}^{K} w_k WCOV_k.$$ (4)

In Equation (4), $w_k$ is the GDP share of component $k$ in the country, while $WCOV_k = \frac{1}{(y)(y_{\cdot k})} \sum_{i=1}^{m} \frac{p_i}{p} (y_i - y)(y_{ik} - y_{\cdot k})$ is the population-weighted coefficient of covariation between total per capita GDP and per capita GDP from component $k$, where $y_{\cdot k}$ is per capita GDP from component $k$ in the country.

Substituting Equations (3) and (4) into Equation (2), we obtain the following bi-dimensional inequality decomposition equation:

$$WCV^2 = \sum_{i=1}^{m} \left(\frac{p_i}{p}\right)\left(\frac{y_i}{y}\right)^2 \sum_{k=1}^{K} w_{ik} WCOV_{ik} + \sum_{k=1}^{K} w_k WCOV_k.$$ (5)

Dividing this equation by $WCV^2$ results in

$$1 = \sum_{i=1}^{m} \left(\frac{p_i}{p}\right)\left(\frac{y_i}{y}\right)^2 \sum_{k=1}^{K} w_{ik} g_{ik} + \sum_{k=1}^{K} w_k g_k = \sum_{i=1}^{m} \sum_{k=1}^{K} c_{ik} + \sum_{k=1}^{K} c_k$$ (6)

where $g_{ik} = \frac{WCOV_{ik}}{WCV^2}$ and $g_k = \frac{WCOV_k}{WCV^2}$. In Equation (6), $c_{ik} = \left(\frac{p_i}{p}\right)\left(\frac{y_i}{y}\right)^2 w_{ik} g_{ik}$ is the contribution of region $i$'s within-region inequality for component $k$ to overall inequality, while $c_k = w_k g_k$ is the contribution of between-region inequality for component $k$ to overall inequality.

In this study, Indonesia was divided into 3 regions; that is, $m = 3$ (see Figure 3). If there are 9 GDP components (9 industrial sectors), then, including components for the between-region inequality, there are $(3 + 1) \times 9 = 36$ components in Equation (6).

## 4. Results

### 4.1. Trends in Inter-Provincial Inequality in Per Capita GDP and $\beta$-Convergence across Provinces for the Period 2010–2020

We first examined the trend in inter-provincial inequality in per capita GDP across 33 provinces for the period 2010–2020 by using the Gini coefficient and the Theil $L$ and $T$ indices (Figure 4). (The Gini coefficient is defined by Gini $= \frac{2}{n\mu} \text{cov}(\boldsymbol{y}, i(\boldsymbol{y}))$, where $n$ is the total number of provinces, $y_i$ is per capita GDP of province $i$, $\mu = \frac{1}{n} \sum_{i=1}^{n} y_i$ is the simple average of per capita GDP, $\boldsymbol{y} = (y_1, y_2, \cdots, y_n)$ is a vector of per capita GDP, and $i(\boldsymbol{y})$ is the ranking of provinces in terms of per capita GDP. The Theil $L$ and $T$ indices are defined, respectively, by $L = \frac{1}{n} \sum_{i=1}^{n} \ln\left(\frac{\mu}{y_i}\right)$ and $T = \frac{1}{n} \sum_{i=1}^{n} \frac{y_i}{\mu} \ln\left(\frac{y_i}{\mu}\right)$. These inequality measures satisfy anonymity, income homogeneity, population homogeneity, and the Pigou–Dalton transfer principle (Anand [9]; Fields [10]).) All inequality measures exhibit a declining trend, implying that inter-provincial inequality in per capita GDP has been decreasing over the study period of 2010–2020. In other words, the provinces exhibit $\sigma$-convergence (Barro and Sala-i-Martin, [24]). To examine which provinces are responsible for the declining inequality, we next performed a $\beta$-convergence analysis across 33 provinces before and during the COVID-19 pandemic (2010–2019 and 2019–2020, respectively).

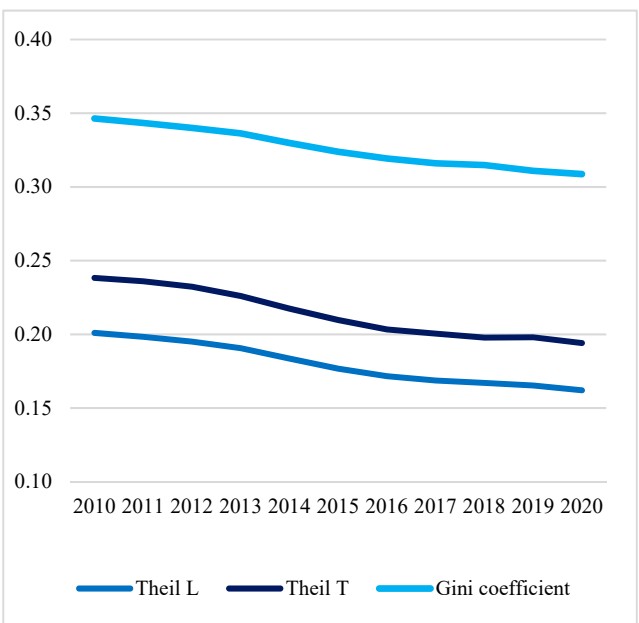

**Figure 4.** Inequality in per capita GDP, 2010–2020. Source: Authors' calculation based on Central Bureau of Statistics [1].

Figure 5 presents per capita GDP by province in 2010. East Kalimantan had the largest per capita GDP, which was followed by Jakarta, Riau, Riau Islands, and West Papua. Except Jakarta, these provinces are resource-rich provinces. On the other hand, East Nusa Tenggara registered the smallest per capita GDP, followed by Maluku, North Maluku, West Sulawesi, and Gorontalo. All these provinces are in region 3 (see Figure 3). Between 2010 and 2019, the Sulawesi provinces performed relatively well in terms of per capita GDP growth. Central Sulawesi recorded the highest growth rate at 8.5%, which was followed by South Sulawesi, West Sulawesi, South East Sulawesi, and Gorontalo. (Central Sulawesi's exceptionally high growth is attributable to the rapid development of the basic metal products sector. In 2020, the sector accounted for 16% of the province's total GDP.) Meanwhile, resource-rich provinces, such as Aceh, Riau, East Kalimantan, and Papua have stagnated. Papua had the smallest per capita GDP growth, which was followed by Riau, East Kalimantan, and Aceh. Their average annual growth rates were below 1%.

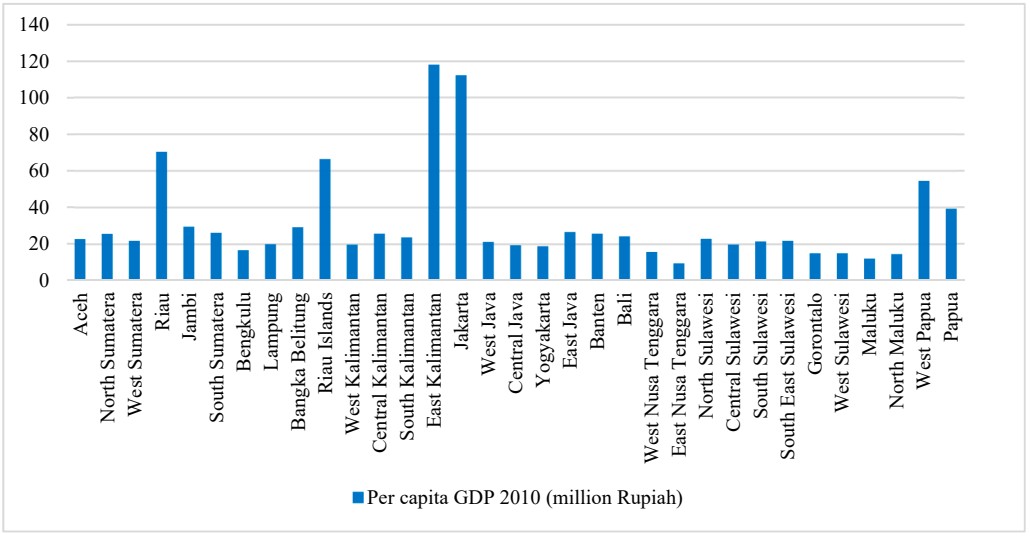

**Figure 5.** Per capita GDP by Province in 2010. Source: Authors' calculation based on Central Bureau of Statistics [1].

Figure 6 exhibits a scatterplot of the average annual growth rate of per capita GDP for 2010–2019 against the natural logarithm of per capita GDP in 2010. There appears to be a negative relationship between these two variables, with a simple correlation coefficient of −0.42. This indicates that there was absolute $\beta$-convergence in a cross-section of 33 provinces; that is, poorer provinces tended to grow faster than richer provinces during the 2010–2019 period. If, in a cross-section of provinces, there is a negative relationship between the initial per capita GDP and subsequent growth without controlling for any conditioning variables, then there is absolute $\beta$-convergence across these provinces (Barro and Sala-i-Martin [24]). On the other hand, if a negative relationship exists between them after controlling for some conditioning variables, then the provinces exhibit conditional $\beta$-convergence.

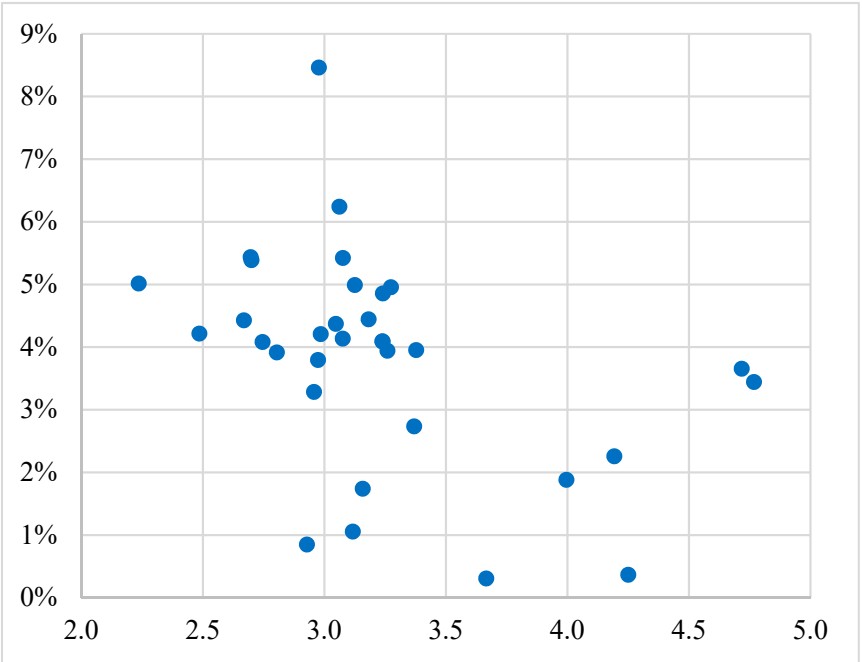

**Figure 6.** Scatterplot of growth rate of per capita GDP for 2010–2019 against log of per capita GDP in 2010. Note: The annual average growth rate of per capita GDP for 2010–2019 is on the vertical axis, while the natural logarithm of per capita GDP in 2010 is on the horizontal axis. Source: Authors' calculation based on Central Bureau of Statistics [1].

In 2019, Jakarta moved to the top position in terms of per capita GDP, followed by East Kalimantan, Riau Islands, and Riau (Figure 7). On the other hand, East Nusa Tenggara, Maluku, and North Maluku were still among the poorest provinces. The COVID-19 pandemic, which started in early 2020, exerted an enormous impact on the Indonesian economy. The country's GDP declined by 2.7% in 2020. However, the pandemic had differential impacts on provincial economies. While tourist destination provinces such as Riau Islands and Bali experienced a large decrease in per capita GDP, some provinces in the eastern part of Indonesia, such as Central Sulawesi and North Maluku, recorded a positive growth. Figure 8 depicts a scatterplot of the growth rate of per capita GDP for 2019–2020 against the natural logarithm of per capita GDP in 2019. No discernible pattern is observed in the relationship between these two variables, indicating that there was no absolute $\beta$-convergence across provinces during the pandemic. This does not, however, rule out the possibility of conditional $\beta$-convergence across these provinces (see footnote 8 for the concept of conditional $\beta$-convergence).

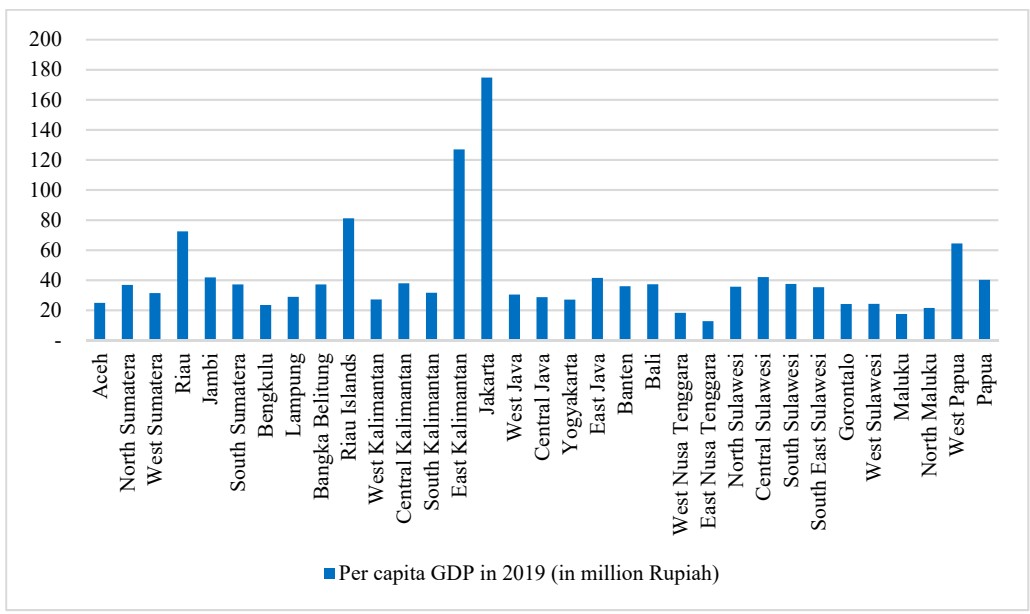

**Figure 7.** Per Capita GDP by Province in 2019. Source: Authors' calculation based on Central Bureau of Statistics [1].

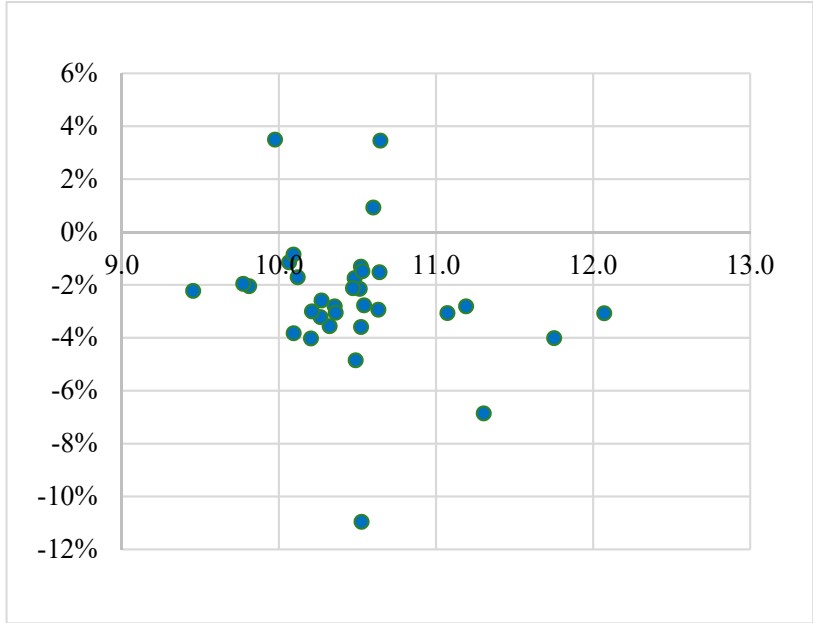

**Figure 8.** Scatterplot of Growth Rate of Per Capita for 2019–2020 against Log of Per Capita GDP in 2019. Note: The annual average growth rate of per capita GDP for 2019–2020 is on the vertical axis, while the natural logarithm of per capita GDP in 2019 is on the horizontal axis. Source: Authors' calculation based on Central Bureau of Statistics [1].

*4.2. Bi-Dimensional Inequality Decomposition Analysis*

4.2.1. Before the Outbreak of the COVID-19 Pandemic

To analyze the effects of changes in industrial and spatial structures on inter-provincial inequality in per capita GDP, we conducted a bi-dimensional inequality decomposition analysis before and after the outbreak of the COVID-19 pandemic. The results for the nine main sectors are shown in Table 2, where the values are the % contributions to overall inter-provincial inequality (see Equations (5) and (6)). These nine main sectors are shown in the first column of Table 1. Using the population-weighted coefficient of variation (*WCV*), Table 3 provides inter-provincial inequalities by industrial sector within each region.

**Table 2.** Bi-dimensional inequality decomposition analysis: 9 main sectors.

| contribution to overall inter-provincial inequality in % | | | | | | | | | | | |
|---|---|---|---|---|---|---|---|---|---|---|---|
| 2010 | 1 | 2 | 3 | 4 | 5 | 6 | 7 | 8 | 9 | Total | GDP Share |
| Total | 1.2 | 19.5 | 16.2 | 0.2 | 12.0 | 15.4 | 7.3 | 17.7 | 10.4 | 100.0 | 100.0 |
| Between-region | 0.6 | 1.1 | 1.2 | 0.0 | 0.2 | 0.3 | 0.1 | 0.1 | −0.2 | 3.5 | |
| Within-region | 0.5 | 18.4 | 15.0 | 0.2 | 11.8 | 15.1 | 7.3 | 17.6 | 10.6 | 96.5 | |
| Region 1 | 2.1 | 17.4 | 9.1 | 0.0 | 2.3 | 1.3 | 0.5 | 0.3 | 0.5 | 33.6 | 31.8 |
| Region 2 | −1.7 | −0.2 | 5.4 | 0.2 | 9.3 | 13.7 | 6.7 | 17.2 | 10.0 | 60.6 | 58.6 |
| Region 3 | 0.1 | 1.2 | 0.4 | 0.0 | 0.2 | 0.1 | 0.1 | 0.1 | 0.1 | 2.3 | 9.6 |
| 2019 | 1 | 2 | 3 | 4 | 5 | 6 | 7 | 8 | 9 | Total | GDP share |
| Total | −0.2 | 9.0 | 11.8 | 0.3 | 11.7 | 18.2 | 13.3 | 23.7 | 12.1 | 100.0 | 100.0 |
| Between-region | −0.2 | 0.0 | 1.3 | 0.0 | 0.1 | 0.6 | 0.2 | 0.4 | −0.1 | 2.3 | |
| Within-region | 0.0 | 9.1 | 10.6 | 0.3 | 11.6 | 17.6 | 13.1 | 23.4 | 12.2 | 97.7 | |
| Region 1 | 1.2 | 8.8 | 5.0 | 0.0 | 1.6 | 0.9 | 0.4 | 0.2 | 0.4 | 18.4 | 29.4 |
| Region 2 | −1.4 | −0.2 | 5.1 | 0.3 | 9.7 | 16.5 | 12.6 | 23.1 | 11.7 | 77.3 | 60.5 |
| Region 3 | 0.2 | 0.4 | 0.5 | 0.0 | 0.3 | 0.2 | 0.1 | 0.1 | 0.2 | 2.1 | 10.1 |
| 2020 | 1 | 2 | 3 | 4 | 5 | 6 | 7 | 8 | 9 | Total | GDP share |
| Total | −0.1 | 8.7 | 11.0 | 0.3 | 11.4 | 16.9 | 14.6 | 24.7 | 12.5 | 100.0 | 100.0 |
| Between-region | −0.2 | −0.1 | 1.1 | 0.0 | 0.1 | 0.5 | 0.2 | 0.3 | −0.1 | 2.0 | |
| Within-region | 0.0 | 8.8 | 9.9 | 0.3 | 11.3 | 16.3 | 14.4 | 24.4 | 12.6 | 98.0 | |
| Region 1 | 1.3 | 8.4 | 5.0 | 0.0 | 1.6 | 0.8 | 0.4 | 0.2 | 0.3 | 18.0 | 29.5 |
| Region 2 | −1.5 | −0.2 | 4.4 | 0.3 | 9.4 | 15.3 | 13.9 | 24.2 | 12.1 | 77.8 | 60.1 |
| Region 3 | 0.3 | 0.5 | 0.6 | 0.0 | 0.3 | 0.2 | 0.1 | 0.1 | 0.2 | 2.2 | 10.4 |

| (Note) | | | |
|---|---|---|---|
| 1 | Agriculture | 6 | Trade/hotel/restaurant |
| 2 | Mining | 7 | Transportation/Information/communication |
| 3 | Manufacturing | 8 | Financial and business services |
| 4 | Electricity/gas/water | 9 | Government and other services |
| 5 | Construction | | |

Source: Authors' calculation based on Central Bureau of Statistics [1].

**Table 3.** Population-weighted coefficient of variation (*WCV*): 9 main sectors.

| 2010 | 1 | 2 | 3 | 4 | 5 | 6 | 7 | 8 | 9 | Total |
|---|---|---|---|---|---|---|---|---|---|---|
| Total | 0.65 | 2.60 | 0.68 | 0.97 | 1.05 | 0.89 | 0.95 | 2.23 | 1.11 | 0.78 |
| Region 1 | 0.46 | 1.75 | 1.02 | 1.41 | 0.69 | 0.33 | 0.45 | 0.32 | 0.30 | 0.74 |
| Region 2 | 0.38 | 0.61 | 0.32 | 0.73 | 1.21 | 0.90 | 1.07 | 2.21 | 1.36 | 0.79 |
| Region 3 | 0.29 | 1.56 | 1.47 | 0.50 | 0.42 | 0.32 | 0.30 | 0.39 | 0.23 | 0.46 |
| 2019 | 1 | 2 | 3 | 4 | 5 | 6 | 7 | 8 | 9 | Total |
| Total | 0.69 | 2.45 | 0.59 | 1.02 | 0.88 | 0.86 | 1.15 | 2.29 | 1.07 | 0.75 |
| Region 1 | 0.44 | 1.72 | 0.90 | 1.27 | 0.61 | 0.30 | 0.44 | 0.29 | 0.32 | 0.58 |
| Region 2 | 0.41 | 0.82 | 0.27 | 0.85 | 1.03 | 0.87 | 1.23 | 2.22 | 1.29 | 0.83 |
| Region 3 | 0.34 | 1.04 | 1.14 | 0.46 | 0.49 | 0.40 | 0.38 | 0.44 | 0.27 | 0.40 |
| 2020 | 1 | 2 | 3 | 4 | 5 | 6 | 7 | 8 | 9 | Total |
| Total | 0.70 | 2.39 | 0.59 | 0.94 | 0.87 | 0.83 | 1.21 | 2.30 | 1.07 | 0.75 |
| Region 1 | 0.45 | 1.71 | 0.91 | 1.16 | 0.59 | 0.30 | 0.43 | 0.28 | 0.33 | 0.57 |
| Region 2 | 0.41 | 0.82 | 0.26 | 0.80 | 1.02 | 0.85 | 1.26 | 2.24 | 1.31 | 0.83 |
| Region 3 | 0.34 | 1.07 | 1.18 | 0.46 | 0.51 | 0.41 | 0.40 | 0.43 | 0.27 | 0.40 |

Note: See Table 4 for the sector classification. Source: Authors' calculation based on Central Bureau of Statistics [1].

**Table 4.** Bi-dimensional inequality decomposition analysis: manufacturing (main sector 3).

| | | | | | Contribution to the sector's overall inter-provincial inequality in % | | | | | | |
|---|---|---|---|---|---|---|---|---|---|---|---|
| 2010 | m1 | m2 | m3 | m4 | m5 | m6 | m7 | m8 | m9 | Total | GDP Share |
| Total | 30.5 | 9.7 | 5.0 | 8.0 | 9.9 | 16.2 | 2.4 | 17.0 | 1.4 | 100.0 | 100.0 |
| Between-region | 0.1 | 4.7 | 2.8 | 1.8 | 2.7 | 3.9 | 0.9 | 3.7 | 0.3 | 20.9 | |
| Within-region | 30.4 | 5.0 | 2.3 | 6.2 | 7.2 | 12.3 | 1.5 | 13.2 | 1.1 | 79.1 | |
| Region 1 | 28.0 | 8.8 | 0.2 | 6.0 | 2.9 | 8.0 | 0.9 | 1.3 | 0.9 | 57.0 | 26.3 |
| Region 2 | −1.1 | −4.1 | 2.1 | 0.0 | 4.1 | 4.3 | 0.6 | 11.9 | 0.2 | 18.0 | 70.2 |
| Region 3 | 3.5 | 0.3 | 0.0 | 0.2 | 0.2 | 0.0 | 0.0 | 0.0 | 0.0 | 4.1 | 3.5 |
| 2019 | m1 | m2 | m3 | m4 | m5 | m6 | m7 | m8 | m9 | Total | GDP share |
| Total | 13.0 | 18.6 | 6.5 | 7.6 | 10.2 | 20.2 | 2.8 | 19.5 | 1.5 | 100.0 | 100.0 |
| Between-region | −0.9 | 6.5 | 4.9 | 2.2 | 3.5 | 3.9 | 1.3 | 5.9 | 0.3 | 27.6 | |
| Within-region | 14.0 | 12.1 | 1.7 | 5.4 | 6.7 | 16.3 | 1.4 | 13.6 | 1.2 | 72.4 | |
| Region 1 | 11.8 | 13.6 | 0.3 | 4.7 | 2.9 | 11.2 | 0.9 | 1.6 | 1.0 | 47.9 | 24.1 |
| Region 2 | −1.1 | −2.0 | 1.4 | 0.5 | 3.4 | 3.7 | 0.5 | 12.0 | 0.3 | 18.7 | 71.2 |
| Region 3 | 3.2 | 0.5 | 0.0 | 0.2 | 0.4 | 1.4 | 0.0 | 0.0 | 0.0 | 5.8 | 4.7 |
| 2020 | m1 | m2 | m3 | m4 | m5 | m6 | m7 | m8 | m9 | Total | GDP share |
| Total | 13.4 | 21.3 | 5.8 | 7.9 | 10.5 | 21.5 | 2.8 | 15.2 | 1.6 | 100.0 | 100.0 |
| Between-region | −1.0 | 6.6 | 4.2 | 2.0 | 3.8 | 2.8 | 1.1 | 4.9 | 0.3 | 24.7 | |
| Within-region | 14.5 | 14.7 | 1.6 | 5.9 | 6.7 | 18.8 | 1.6 | 10.3 | 1.3 | 75.3 | |
| Region 1 | 11.7 | 15.3 | 0.3 | 5.1 | 3.0 | 12.6 | 1.2 | 1.7 | 1.0 | 51.8 | 24.7 |
| Region 2 | −0.8 | −1.1 | 1.2 | 0.6 | 3.3 | 3.4 | 0.5 | 8.7 | 0.3 | 16.1 | 70.1 |
| Region 3 | 3.5 | 0.5 | 0.0 | 0.3 | 0.4 | 2.7 | 0.0 | 0.0 | 0.0 | 7.4 | 5.1 |

| | (Note) | | | |
|---|---|---|---|---|
| m1 | Coal and refined petroleum products | | m6 | Basic metals and fabricated metal products |
| m2 | Food, tobacco, and beverages | | m7 | Machinery and equipment |
| m3 | Textiles, wearing apparel, and leather products | | m8 | Transport equipment |
| m4 | Wood products, furniture, and paper products | | m9 | Other manufacturing |
| m5 | Chemical products, rubber products, and other non-metallic mineral products | | | |

Source: Authors' calculation based on Central Bureau of Statistics [1].

Some major changes can be observed between 2010 and 2019. First, region 2 raised its contribution to overall inter-provincial inequality conspicuously from 60.6% to 77.3%. The transportation and IC sector and the financial and business services sector were mainly responsible for this increase; the combined contribution of these two sectors rose from 23.9% to 35.7%. We should note that the transportation and IC sector not only raised its inter-provincial inequality (Table 3) but also its GDP share in region 2. On the other hand, the financial and business services sector raised its GDP share, though its inter-provincial inequality remained constant at a high level in region 2 (Table 3). Second, region 1 lowered its contribution from 33.6% to 18.4%. The main contributor was the mining sector. While the mining sector's inter-provincial inequality remained constant at a high level (Table 3), its GDP share declined notably, from 22.6% to 17.6% in region 1. Thus, region 1's mining sector reduced its contribution substantially, from 17.4% to 8.8%. Another contributor is region 1's manufacturing sector. Unlike the mining sector, the manufacturing sector lowered its inter-provincial inequality (Table 3). Though its GDP share remained almost constant in region 1, its contribution to overall inter-provincial inequality declined from 9.1% to 5.0%.

Since the manufacturing and services sectors played an important role in determining inter-provincial inequality in per capita GDP, we next conducted bi-dimensional inequality decomposition analyses for (1) manufacturing subsectors; (2) trade, transportation, and IC subsectors; and (3) finance, business, and government services subsectors. These subsectors are shown in the second column in Table 1. The results are presented in Tables 4–6,

respectively. On the other hand, Tables 7–9 provide inter-provincial inequalities within each region for (1), (2), and (3), respectively.

**Table 5.** Bi-dimensional inequality decomposition analysis: trade, transportation, information and communication (main sectors 6 and 7).

| Contribution to the sector's overall inter-provincial inequality in % | | | | | | | | | | | |
|---|---|---|---|---|---|---|---|---|---|---|---|
| 2010 | t1 | t2 | t3 | t4 | t5 | t6 | t7 | t8 | t9 | Total | GDP Share |
| Total | 51.1 | 17.0 | 0.1 | 2.6 | 1.3 | 0.0 | 0.0 | 3.6 | 24.4 | 100.0 | 100.0 |
| Between-region | 3.4 | 1.7 | 0.0 | 0.1 | −0.1 | −0.1 | −0.1 | 0.2 | 1.2 | 6.3 | |
| Within-region | 47.7 | 15.3 | 0.1 | 2.5 | 1.4 | 0.0 | 0.1 | 3.4 | 23.2 | 93.7 | |
| Region 1 | 1.2 | 0.2 | 0.0 | 0.1 | 0.1 | 0.0 | 0.2 | 0.2 | 0.2 | 2.1 | 22.6 |
| Region 2 | 46.2 | 15.1 | 0.1 | 2.4 | 1.2 | 0.0 | −0.1 | 3.2 | 22.9 | 91.1 | 69.5 |
| Region 3 | 0.3 | 0.0 | 0.0 | 0.0 | 0.0 | 0.0 | 0.0 | 0.0 | 0.1 | 0.5 | 7.9 |
| 2019 | t1 | t2 | t3 | t4 | t5 | t6 | t7 | t8 | t9 | Total | GDP share |
| Total | 42.6 | 15.1 | 0.1 | 3.0 | 0.7 | 0.0 | 1.2 | 4.0 | 33.3 | 100.0 | 100.0 |
| Between-region | 2.7 | 1.7 | 0.0 | 0.2 | −0.1 | −0.1 | −0.1 | 0.2 | 1.9 | 6.5 | |
| Within-region | 39.9 | 13.3 | 0.1 | 2.8 | 0.9 | 0.0 | 1.3 | 3.8 | 31.4 | 93.5 | |
| Region 1 | 0.7 | 0.1 | 0.0 | 0.1 | 0.1 | 0.0 | 0.1 | 0.2 | 0.2 | 1.4 | 21.8 |
| Region 2 | 38.7 | 13.2 | 0.1 | 2.7 | 0.8 | 0.0 | 1.1 | 3.6 | 31.0 | 91.2 | 69.6 |
| Region 3 | 0.5 | 0.1 | 0.0 | 0.1 | 0.0 | 0.0 | 0.0 | 0.0 | 0.2 | 0.8 | 8.6 |
| 2020 | t1 | t2 | t3 | t4 | t5 | t6 | t7 | t8 | t9 | Total | GDP share |
| Total | 40.7 | 12.7 | 0.1 | 3.1 | 0.7 | 0.0 | 0.6 | 4.3 | 37.8 | 100.0 | 100.0 |
| Between-region | 2.6 | 1.6 | 0.0 | 0.2 | −0.1 | −0.1 | −0.1 | 0.2 | 2.4 | 6.7 | |
| Within-region | 38.1 | 11.1 | 0.1 | 2.9 | 0.8 | 0.0 | 0.7 | 4.1 | 35.5 | 93.3 | |
| Region 1 | 0.7 | 0.1 | 0.0 | 0.1 | 0.1 | 0.0 | 0.1 | 0.2 | 0.2 | 1.4 | 21.6 |
| Region 2 | 36.9 | 11.0 | 0.1 | 2.8 | 0.7 | 0.0 | 0.6 | 3.9 | 35.0 | 91.1 | 69.9 |
| Region 3 | 0.5 | 0.1 | 0.0 | 0.0 | 0.0 | 0.0 | 0.0 | 0.0 | 0.2 | 0.9 | 8.5 |
| (Note) | | | | | | | | | | | |
| t1 | Wholesale and retail trade | | | | | t6 | River and lake transportation | | | | |
| t2 | Hotels and restaurants | | | | | t7 | Air transportation | | | | |
| t3 | Railway transportation | | | | | t8 | Support services for transportation | | | | |
| t4 | Road transportation | | | | | t9 | Information and communication | | | | |
| t5 | Sea transportation | | | | | | | | | | |

Source: Authors' calculation based on Central Bureau of Statistics [1].

**Table 6.** Bi-dimensional inequality decomposition analysis: finance, business and government services (main sectors 8 and 9).

| Contribution to the sector's overall inter-provincial inequality in % | | | | | | | | | | | |
|---|---|---|---|---|---|---|---|---|---|---|---|
| 2010 | s1 | s2 | s3 | s4 | s5 | s6 | s7 | s8 | s9 | Total | GDP Share |
| Total | 13.9 | 9.2 | 3.9 | 18.0 | 18.3 | 12.4 | 12.8 | 3.8 | 7.7 | 100.0 | 100.0 |
| Between-region | 0.3 | 0.2 | 0.1 | 0.3 | 0.4 | −0.2 | 0.2 | 0.0 | 0.2 | 1.4 | |
| Within-region | 13.6 | 9.0 | 3.8 | 17.7 | 17.9 | 12.5 | 12.6 | 3.8 | 7.5 | 98.6 | |
| Region 1 | 0.0 | 0.0 | 0.0 | 0.0 | 0.0 | 0.1 | 0.1 | 0.0 | 0.0 | 0.4 | 20.5 |
| Region 2 | 13.5 | 9.0 | 3.8 | 17.7 | 17.9 | 12.3 | 12.5 | 3.8 | 7.5 | 98.0 | 68.7 |
| Region 3 | 0.0 | 0.0 | 0.0 | 0.0 | 0.0 | 0.1 | 0.0 | 0.0 | 0.0 | 0.2 | 10.9 |
| 2019 | s1 | s2 | s3 | s4 | s5 | s6 | s7 | s8 | s9 | Total | GDP share |
| Total | 14.9 | 9.3 | 3.7 | 16.1 | 22.4 | 8.9 | 11.0 | 4.0 | 9.7 | 100.0 | 100.0 |
| Between-region | 0.4 | 0.2 | 0.1 | 0.3 | 0.6 | −0.2 | 0.3 | 0.0 | 0.3 | 1.9 | |
| Within-region | 14.5 | 9.1 | 3.6 | 15.8 | 21.8 | 9.2 | 10.7 | 4.0 | 9.4 | 98.1 | |
| Region 1 | 0.0 | 0.0 | 0.0 | 0.0 | 0.0 | 0.1 | 0.1 | 0.0 | 0.0 | 0.3 | 19.7 |
| Region 2 | 14.5 | 9.1 | 3.6 | 15.7 | 21.8 | 9.0 | 10.6 | 4.0 | 9.4 | 97.5 | 68.9 |
| Region 3 | 0.0 | 0.0 | 0.0 | 0.1 | 0.0 | 0.1 | 0.0 | 0.0 | 0.0 | 0.3 | 11.4 |

**Table 6.** *Cont.*

| 2020 | s1 | s2 | s3 | s4 | s5 | s6 | s7 | s8 | s9 | Total | GDP share |
|---|---|---|---|---|---|---|---|---|---|---|---|
| Total | 15.6 | 9.6 | 3.6 | 16.2 | 21.7 | 8.2 | 11.1 | 4.9 | 9.3 | 100.0 | 100.0 |
| Between-region | 0.4 | 0.2 | 0.1 | 0.3 | 0.5 | −0.3 | 0.3 | 0.0 | 0.3 | 1.8 | |
| Within-region | 15.2 | 9.3 | 3.5 | 15.9 | 21.1 | 8.5 | 10.8 | 4.8 | 8.9 | 98.2 | |
| Region 1 | 0.0 | 0.0 | 0.0 | 0.0 | 0.0 | 0.1 | 0.1 | 0.0 | 0.0 | 0.3 | 19.7 |
| Region 2 | 15.2 | 9.3 | 3.5 | 15.8 | 21.1 | 8.3 | 10.7 | 4.8 | 8.9 | 97.6 | 68.7 |
| Region 3 | 0.0 | 0.0 | 0.0 | 0.1 | 0.0 | 0.1 | 0.0 | 0.0 | 0.0 | 0.3 | 11.5 |

| (Note) | | | | | |
|---|---|---|---|---|---|
| s1 | Financial intermediary services | | s6 | Public administration | |
| s2 | Insurance and pension fund | | s7 | Education services | |
| s3 | Other financial services | | s8 | Health services | |
| s4 | Real estate | | s9 | Other services | |
| s5 | Business services | | | | |

Source: Authors' calculation based on Central Bureau of Statistics [1].

**Table 7.** Population-weighted coefficient of variation (*WCV*): manufacturing (main sector 3).

| 2010 | m1 | m2 | m3 | m4 | m5 | m6 | m7 | m8 | m9 | Total |
|---|---|---|---|---|---|---|---|---|---|---|
| Total | 3.39 | 0.86 | 1.15 | 0.93 | 0.71 | 1.62 | 1.85 | 2.47 | 1.97 | 0.68 |
| Region 1 | 2.84 | 0.96 | 2.26 | 1.43 | 0.53 | 3.71 | 4.27 | 4.53 | 4.56 | 1.02 |
| Region 2 | 1.10 | 0.63 | 0.68 | 0.50 | 0.62 | 0.81 | 1.32 | 1.86 | 0.45 | 0.32 |
| Region 3 | 6.48 | 0.78 | 0.52 | 1.16 | 1.47 | 1.84 | 5.52 | 1.02 | 0.46 | 1.47 |
| **2019** | **m1** | **m2** | **m3** | **m4** | **m5** | **m6** | **m7** | **m8** | **m9** | **Total** |
| Total | 3.00 | 0.85 | 1.12 | 0.86 | 0.69 | 1.64 | 1.84 | 2.33 | 2.06 | 0.59 |
| Region 1 | 2.69 | 0.93 | 2.29 | 1.37 | 0.70 | 3.91 | 3.99 | 4.40 | 4.29 | 0.90 |
| Region 2 | 1.16 | 0.66 | 0.63 | 0.49 | 0.52 | 0.73 | 1.31 | 1.71 | 0.54 | 0.27 |
| Region 3 | 5.23 | 0.83 | 0.44 | 1.04 | 1.43 | 2.57 | 5.55 | 1.05 | 0.33 | 1.14 |
| **2020** | **m1** | **m2** | **m3** | **m4** | **m5** | **m6** | **m7** | **m8** | **m9** | **Total** |
| Total | 3.10 | 0.86 | 1.11 | 0.87 | 0.70 | 1.73 | 1.86 | 2.24 | 1.98 | 0.59 |
| Region 1 | 2.66 | 0.96 | 2.32 | 1.39 | 0.73 | 3.97 | 4.26 | 4.33 | 4.24 | 0.91 |
| Region 2 | 1.08 | 0.65 | 0.62 | 0.47 | 0.52 | 0.73 | 1.30 | 1.64 | 0.59 | 0.26 |
| Region 3 | 5.29 | 0.84 | 0.43 | 1.09 | 1.43 | 2.62 | 5.56 | 1.06 | 0.33 | 1.18 |

Note: See Table 6 for the sector classification. Source: Authors' calculation based on Central Bureau of Statistics [1].

**Table 8.** Population-weighted coefficient of variation (*WCV*): trade, transportation, information and communication (main sectors 6 and 7).

| 2010 | t1 | t2 | t3 | t4 | t5 | t6 | t7 | t8 | t9 | Total |
|---|---|---|---|---|---|---|---|---|---|---|
| Total | 0.83 | 1.36 | 1.27 | 0.60 | 1.42 | 2.24 | 1.21 | 1.48 | 1.38 | 0.90 |
| Region 1 | 0.33 | 0.53 | 1.34 | 0.68 | 0.99 | 1.53 | 0.97 | 1.36 | 0.36 | 0.30 |
| Region 2 | 0.88 | 1.14 | 1.02 | 0.55 | 1.82 | 3.76 | 1.56 | 1.44 | 1.41 | 0.95 |
| Region 3 | 0.31 | 0.56 | 0.00 | 0.52 | 0.95 | 0.95 | 0.47 | 0.51 | 0.40 | 0.29 |
| **2019** | **t1** | **t2** | **t3** | **t4** | **t5** | **t6** | **t7** | **t8** | **t9** | **Total** |
| Total | 0.79 | 1.30 | 1.18 | 0.61 | 1.34 | 2.28 | 1.08 | 1.61 | 1.55 | 0.95 |
| Region 1 | 0.30 | 0.51 | 1.38 | 0.69 | 1.07 | 1.45 | 0.86 | 1.30 | 0.38 | 0.27 |
| Region 2 | 0.85 | 1.08 | 0.90 | 0.56 | 1.46 | 4.14 | 1.33 | 1.58 | 1.51 | 0.99 |
| Region 3 | 0.40 | 0.61 | 0.00 | 0.54 | 1.06 | 0.95 | 0.50 | 0.59 | 0.52 | 0.38 |
| **2020** | **t1** | **t2** | **t3** | **t4** | **t5** | **t6** | **t7** | **t8** | **t9** | **Total** |
| Total | 0.78 | 1.21 | 1.49 | 0.65 | 1.41 | 2.35 | 1.10 | 1.74 | 1.52 | 0.96 |
| Region 1 | 0.30 | 0.45 | 1.44 | 0.68 | 1.12 | 1.51 | 1.02 | 1.40 | 0.38 | 0.28 |
| Region 2 | 0.84 | 0.99 | 1.28 | 0.62 | 1.54 | 4.28 | 1.34 | 1.70 | 1.46 | 1.00 |
| Region 3 | 0.41 | 0.61 | 0.00 | 0.55 | 1.12 | 0.92 | 0.50 | 0.60 | 0.53 | 0.40 |

Note: See Table 7 for the sector classification. Source: Authors' calculation based on Central Bureau of Statistics [1].

**Table 9.** Population-weighted coefficient of variation (*WCV*): finance, business and government services (main sectors 8 and 9).

| 2010 | s1 | s2 | s3 | s4 | s5 | s6 | s7 | s8 | s9 | Total |
|---|---|---|---|---|---|---|---|---|---|---|
| Total | 1.77 | 3.42 | 1.94 | 1.74 | 3.51 | 0.96 | 1.22 | 1.14 | 1.43 | 1.62 |
| Region 1 | 0.34 | 1.42 | 0.66 | 0.29 | 0.87 | 0.33 | 0.34 | 0.48 | 0.40 | 0.26 |
| Region 2 | 1.81 | 2.98 | 2.07 | 1.88 | 2.94 | 1.30 | 1.40 | 1.47 | 1.36 | 1.80 |
| Region 3 | 0.33 | 0.85 | 0.53 | 0.47 | 1.16 | 0.42 | 0.28 | 0.44 | 0.19 | 0.24 |
| **2019** | **s1** | **s2** | **s3** | **s4** | **s5** | **s6** | **s7** | **s8** | **s9** | **Total** |
| Total | 1.88 | 3.43 | 1.81 | 1.62 | 3.70 | 0.92 | 1.02 | 1.10 | 1.66 | 1.64 |
| Region 1 | 0.28 | 1.27 | 0.62 | 0.32 | 0.86 | 0.33 | 0.40 | 0.46 | 0.40 | 0.25 |
| Region 2 | 1.89 | 2.95 | 1.91 | 1.75 | 3.03 | 1.28 | 1.14 | 1.37 | 1.54 | 1.79 |
| Region 3 | 0.27 | 0.79 | 0.73 | 0.56 | 1.15 | 0.49 | 0.35 | 0.50 | 0.31 | 0.29 |
| **2020** | **s1** | **s2** | **s3** | **s4** | **s5** | **s6** | **s7** | **s8** | **s9** | **Total** |
| Total | 1.92 | 3.46 | 1.80 | 1.63 | 3.77 | 0.89 | 1.02 | 1.21 | 1.71 | 1.65 |
| Region 1 | 0.28 | 1.22 | 0.62 | 0.33 | 0.86 | 0.32 | 0.41 | 0.46 | 0.42 | 0.25 |
| Region 2 | 1.94 | 2.96 | 1.90 | 1.75 | 3.07 | 1.25 | 1.13 | 1.50 | 1.57 | 1.80 |
| Region 3 | 0.26 | 0.80 | 0.75 | 0.57 | 1.17 | 0.49 | 0.37 | 0.52 | 0.30 | 0.28 |

Note: See Table 8 for the sector classification. Source: Authors' calculation based on Central Bureau of Statistics [1].

Within the manufacturing sector, region 1 was the main contributor to the sector's overall inter-provincial inequality (Table 4), but it lowered its contribution in the 2010–2019 period (from 57.0 to 47.9%). The coal and refined petroleum products sector was mainly responsible for this decrease. Its inter-provincial inequality was very high in region 1 due to the very uneven spatial distribution of the sector's activities (Table 7). But, its GDP share declined substantially in region 1, from 23.6% to 15.0%, resulting in a large reduction in the contribution to manufacturing's overall inter-provincial inequality (from 28.0% to 11.8%). In region 1, the food, tobacco, and beverages sector raised its contribution from 8.8% to 13.6% in the 2010–2019 period owing to the increase in its GDP share. But, this could not offset the large reduction in the contribution of the coal and refined petroleum products sector. From these observations, the decrease in the manufacturing sector's contribution to overall inter-provincial inequality was attributable mainly to the declining role of the coal and refined petroleum products sector in region 1 (see Table 2).

Within the trade, transportation, and IC sector, region 2 dominated, accounting for 91% of the sector's overall inter-provincial inequality (Table 5). The IC sector raised its contribution substantially from 24.4% to 33.3% in the 2010–2019 period; but, region 2's IC sector was mostly responsible for this increase. In region 2, the IC sector had a very large inter-provincial inequality because its activities were concentrated in a few major cities on the island of Java, such as Jakarta and Surabaya (Table 8). It grew very rapidly, at an annual average rate of 9.9%; its GDP share increased from 17.0% to 22.4% in region 2. This, together with rising inequality, caused the contribution of region 2's IC sector to increase from 22.9% to 31.0%. We should note that region 2 accounted for 75% of total GDP generated by the IC sector in 2019. On the other hand, the wholesale and retail trade sector reduced its contribution from 46.2% to 38.7% in region 2 (Table 5). As discussed before, the transportation and IC sector raised its contribution to overall inter-provincial inequality (Table 2). But, this was attributable mainly to the rapid growth in the IC sector in region 2.

Like the trade, transportation, and IC sector, region 2 dominated in the finance, business and government services sector, accounting for 98% of the sector's overall inter-provincial inequality (Table 6). The business services sector raised its contribution from 18.3% to 22.4% in the 2010–2019 period; but, region 2's business services sector was mostly responsible for this increase. Reflecting the very uneven spatial distribution of knowledge-intensive business services, the sector had an exceptionally high inter-provincial inequality in region 2 (Table 9) (Jakarta dominated in the business services sector by accounting for half of region 2's total GDP). It grew very rapidly at an annual average rate of 8.1%. While its GDP share increased only slightly, from 11.2% to 13.2% in region 2, the contribution

of region 2's business services sector rose from 17.9% to 21.8%. The financial sector also increased its contribution, from 13.9% to 14.9%. Like the business services sector, region 2's financial sector was responsible for this increase. We should note that region 2 accounted for 91.4% of total GDP generated by the business services sector and 74.3% of total GDP generated by the financial sector in 2019. As discussed before, the finance, business, and government services sector raised its contribution to overall inter-provincial inequality (Table 2). But, this was attributable mainly to a rapid growth of the business services sector in region 2.

### 4.2.2. After the Outbreak of COVID-19

The COVID-19 pandemic had differential impacts on industrial sectors. How have structural changes caused by the pandemic affected inter-provincial inequality in per capita GDP? To answer this question, we conducted bi-dimensional inequality decomposition analyses for the year 2020. The results are presented in Tables 2–6. We can observe some major changes between 2019 and 2020.

Within the trade, transportation, and IC sector, the hotel and restaurant sector lowered its contribution substantially, from 15.1% to 12.7% (Table 5). Region 2's hotel and restaurant sector was mainly responsible for this decrease. In region 2, the sector contracted substantially due to the pandemic (−13.5%), and its GDP share declined from 14.7% to 13.4%. On the other hand, the IC sector was not affected by the pandemic and raised its contribution from 33.3% to 37.8% (Table 5). Region 2's IC sector was mainly responsible for this increase. In region 2, the IC sector grew very rapidly even during the pandemic (at 13.5%), and its GDP share increased notably from 22.4% to 26.5%.

Within the finance, business, and government services sector, the financial services sector raised its contribution from 14.9% to 15.6% (Table 6). Region 2's financial services sector was wholly responsible for this increase. In region 2, the sector grew at 3.5% and its GDP share increased, though slightly, from 14.1% to 14.5%. The health services sector also increased its contribution from 4.0% to 4.9%. Like the financial services sector, region 2's health services sector was wholly responsible for this increase. In region 2, the sector grew very rapidly (at 10.3%), and its GDP share rose from 5.3% to 5.9%. We should note that the health services sector grew rapidly in all regions owing to increasing demands for health services during the pandemic. During the pandemic, most education services were provided using online remote teaching; thus, the education services sector was relatively unaffected by the pandemic. Though the growth rate was much smaller than the health services sector's, the sector grew at 2.6%, and its contribution remained constant in 2020 (see Table 8).

On the other hand, the business services sector reduced its contribution from 22.4% to 21.7% (Table 6). Region 2's business services sector was responsible for this decrease. Business services were concentrated in a few cities in region 2; Jakarta accounted for three-quarters of the total GDP generated by the business services sector in 2020. The sector had a very high inter-provincial inequality in region 2 (Table 9). Unlike the IC sector, the business services sector was affected by the pandemic. In region 2, the sector contracted by 3.4%, and its GDP share declined from 13.2% to 12.7%. We should note that the public administration sector also lowered its contribution as it contracted by 1.4%; but, its contribution was not large (8.2% in 2020) because it had a much smaller inter-provincial inequality than the business services sector (Table 9).

Tourism sectors were hit very hard by the COVID-19 pandemic. In addition to the hotel and restaurant sector, the textile and apparel, transport equipment, and air transportation sectors contracted substantially; their GDP growth rates were, respectively, −9.7%, −15.3%, and −15.8% in 2020 (among transportation sectors, air transportation was hit hardest due to restricted movement between Indonesian islands and between countries). They lowered their contributions to overall inter-provincial inequality, though only slightly. Provinces with a higher GDP share of tourism sectors, such as Bali and Riau Islands, recorded a large negative growth. As discussed above, the IC and financial services sectors were not affected

by the pandemic. These two sectors had a high inequality in per capita GDP, particularly in region 2; using the *WCV*, their inequalities were 1.5 and 1.9 in region 2, respectively (Tables 8 and 9). Thus, they played an increasingly important role in determining inter-provincial inequality in per capita GDP.

## 5. Concluding Remarks

Based on provincial GDP across industrial sectors, this study investigated how structural changes caused by the pandemic have affected the determinants of inter-provincial inequality in per capita GDP by conducting a bi-dimensional inequality decomposition analysis.

The major findings are summarized as follows: First, inter-provincial inequality in per capita GDP, as measured by the Gini coefficient and the Theil indices, has been decreasing over the study period of 2010–2020. Before the COVID-19 pandemic (2010–2019), relatively poor Sulawesi provinces grew faster than other provinces, while resource-rich provinces (such as Aceh, Riau, East Kalimantan, and Papua) were stagnant, indicating there was absolute $\beta$-convergence across Indonesian provinces. In contrast, no absolute $\beta$-convergence was observed across these provinces during the pandemic (2019–2020), though this does not rule out the possibility of conditional $\beta$-convergence.

Second, the results of a bi-dimensional inequality decomposition analysis show that before the outbreak of COVID-19, the Java-Bali region (region 2) raised its contribution to overall inter-provincial inequality from 61% to 77%. The IC (information and communication), financial services, and business services sectors were mainly responsible for this increase; these three sectors had very large inter-provincial inequalities and grew very rapidly in region 2. On the other hand, the Sumatra and Kalimantan region (region 1) lowered its contribution to overall inequality from 34% to 18%. The main contributors were the mining sector and the coal and refined petroleum products sector. While these two sectors had very large inter-provincial inequalities, they lowered their GDP shares substantially in region 1, resulting in a large reduction in their contributions to overall inequality.

Third, after the outbreak of COVID-19, the hotel and restaurant sector, one of the tourism sectors, lowered its contribution to overall inter-provincial inequality prominently. Region 2 was responsible for this decrease, where the sector contracted substantially (−14%), and its GDP share declined. Other tourism sectors, such as textile and apparel, transport equipment, and air transportation, also contracted substantially and lowered their contributions. In contrast, the IC and financial services sectors were not affected by the pandemic and raised their contributions to overall inequality. These two sectors had high inter-provincial inequality, particularly in region 2. They have played an increasingly important role in determining overall inequality. Owing to increasing demands for health services, the health services sector grew very rapidly, but its contribution to overall inequality was not large. On the other hand, the business services sector was severely affected by the pandemic. It experienced a negative growth and lowered its contribution. However, with its very large inter-provincial inequality in region 2, it still serves as one of the main contributors to overall inequality.

The COVID-19 pandemic has exerted an enormous impact on the Indonesian economy. In 2020, the country contracted by 2.7% in real GDP. But, the impact has been spatially heterogeneous. Many provinces, particularly those relying on tourism, experienced large negative growth, while some poorer provinces in the eastern part of Indonesia escaped severe economic downturn. This finding is consistent with those of a study conducted by Gibson and Olivia [17], in which richer provinces with lower initial poverty headcount ratios, such as Bali and Riau Islands, tended to exhibit a larger increase in poverty incidence, while poorer provinces in the eastern part of Indonesia, such as Central Sulawesi, Papua, and North Maluku, tended to experience a smaller increase in poverty incidence.

When Indonesia recovers from the pandemic, an important policy question is whether inter-provincial inequality in per capita GDP will further decrease or not. Another important policy question is which industrial sectors will serve to determine inter-provincial inequality. It is likely that the mining sector and the coal and refined petroleum products

sector will further reduce their significance as their GDP shares will decrease. It is also likely that the tourism sector will regain its position as an important determinant of inter-provincial inequality. However, the most important sectors in determining inter-provincial inequality will be the IC, financial, and business services sectors. With the rapid advancement of IC, financial, and e-business technologies, the roles of these high-inequality services sectors are likely to increase in determining inter-provincial inequality, unless policies that could facilitate spatial dispersion of these services and activities are implemented. On the other hand, the manufacturing sector is likely to reduce inter-provincial inequality as the GDP share of inequality-reducing manufacturing sectors such as food processing will increase.

While our study provides valuable insights into the initial impacts of the COVID-19 pandemic on inter-provincial income inequality, it is not without limitations. First, the bi-dimensional inequality decomposition method that we employed is descriptive. Thus, in future research, we plan to conduct an econometric analysis using panel data for 33 provinces to assess the effects of the pandemic on provincial economies. Second, due to the unavailability of provincial GDP data for the 52 industrial sectors for the period after 2021, we could not analyze the extended effects of the COVID-19 pandemic on inter-provincial income inequality in a comparable analytical framework. Thus, we plan to conduct further research to investigate the longer-term effects using provincial GDP data for the 52 industrial sectors for the period from 2019 to 2023.

**Author Contributions:** Conceptualization, A.S.A.; Methodology, T.A.; Writing—original draft, T.A.; Writing—review & editing, A.S.A. All authors have read and agreed to the published version of the manuscript.

**Funding:** Akita is grateful to the Japan Society for the Promotion of Science (Grant-in-Aid for Scientific Research 18K01589 and 23K01409).

**Institutional Review Board Statement:** Not applicable.

**Informed Consent Statement:** Not applicable.

**Data Availability Statement:** Restrictions apply to the availability of these data. Data was obtained from the Central Bureau of Statistics (CBS) and are available from the authors with the permission of the CBS.

**Conflicts of Interest:** The authors declare no conflict of interest.

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
