# Peer review of "The Initial Impacts of the COVID-19 Pandemic on Regional Economies in Indonesia: Structural Changes and Regional Income Inequality"

_sustainability, doi:10.3390/su151813709_

Round 1

Reviewer 1 Report

please see attached file. 

No need.

Author Response

Comment 1

Line 99-101 stated as “Our study also differs from theirs in that it uses a 52-sector classification, while Akita, Kurniawan and Miyata (2011) and Alisjahbana and Akita (2020) used, respectively, 9-sector and 33-sector classifications (see Table 1 for the sector classifications). Thus, our study can analyze the impact of structural changes on inter-provincial inequality in more detail.”. However, this statement is incomplete or not enough explanatory to claim your contribution. Each point of the contribution of the study should be based on strong justification(s). The author (s) should revise it.

Response

We added the following sentences in the introduction. “With 9-sector and 33-sector classifications, we are not able to analyze structural changes within the manufacturing sector. In the 33-sector classification, the manufacturing sector consists of two subsectors: oil and gas manufacturing and non-oil and non-gas manufacturing, while it contains 16 subsectors in the 52-classification (see Table 1). The manufacturing sector has served as the engine of growth and played an important role in regional economies. With 16 subsectors, therefore, our study provides much better insights into the role of the manufacturing sector in inter-provincial income inequality.”

Comment 2

Please provide a summary of recent literature on the topic in chronological order.

Response

We provided the following footnote in the introduction after “To the best of our knowledge, only a few studies have investigated the impact of the COVID-19 pandemic on the distribution of income in Indonesia.” (Footnote: “Following the completion of this paper, several additional articles addressing the impacts of the COVID-19 on income inequality have been published. They include Brata, Pramudy, Astuti, Rahayu and Heron (2021), Suryahadi, Izzati and Yumna (2021), and Novianti and Panjaitan (2022). But, they differ from our study in both methodology and data.”)

Comment 3

The results are meaningful but robustness or sensitivity analysis should be carried out to authenticate the results.

Response

Since we conducted a descriptive analysis, not an econometric analysis, we are neither able to do a robustness check nor a sensitivity analysis.

Comment 4

Discussion of main results with contextualization i.e. consistency or contradiction with prior studies is very short. The author (s) needs to expand it.

Response

In the conclusion, we added the following sentence after “Many provinces, particularly those relying on tourism experienced a large negative growth, while some poorer provinces in the eastern part of Indonesia escaped from a severe economic downturn.”

“This finding is consistent with the study conducted by Gibson and Olivia (2020), in which richer provinces with lower initial poverty headcount ratios, such as Bali and Riau Islands, tend to exhibit a larger increase in poverty incidence while poorer provinces in the eastern part of Indonesia, such as Papua and North Maluku, tend to experience a smaller increase in poverty incidence.”

Comment 5

The policy implications should be drawn from obtained results and should be precisely linked with your study findings.

Response

Policy implications provided in the conclusion are in line with our findings from a bi-dimensional inequality decomposition analysis.

Reviewer 2 Report

The article under the title "The Initial Impacts of the COVID-19 Pandemic on Regional Economies in Indonesia: Structural Changes and Regional Income Inequality" fits in with the journal's theme. Nevertheless, it contains numerous editing as well as substantive errors. The main objections include: (1) the absence in the abstract of the main aim, the research methods as well as the contribution of the conducted study to the development of theory and/or practice; (2) the inadequate structure of the introduction (the authors should start by identifying the research gaps that become the basis for defining the main objective and the research hypotheses); (3) the absence of a literature review section (the authors could have used an excerpt from the current introduction and expanded it with the research results of other authors); (4) the absence in the description of the methods of justification for the chosen research period (2010-2020?); (5) too superficial discussion of the results (the authors should confront the obtained results with the results of other authors, as well as refer to the research questions and research hypotheses); (6) misconstrued summary (the authors should refer to the research gaps, main aim and research hypotheses, then they should present the most important conclusions, indicate what were the limitations of the conducted study and indicate future research directions); (7) the article does not meet the editorial requirements of the journal.

The article does not point out any significant linguistic errors.

Author Response

Comment 1

Absence in the abstract of the main aim, the research methods as well as the contribution of the conducted study to the development of theory and/or practice

Response

We included, in the abstract, the main objective and the method as follows. “Based on provincial GDP by industrial sectors, this study examines how structural changes caused by the pandemic have affected the determinants of inter-provincial inequality in Indonesia by conducting a bi-dimensional inequality decomposition analysis.” The method we employed is the bi-dimensional inequality decomposition method developed by Akita, Kurniawan and Miyata (2011).

Comment 2

Inadequate structure of the introduction (the authors should start by identifying the research gaps that become the basis for defining the main objective and the research hypotheses)

Response

In the introduction, we stated that “the COVID-19 pandemic hit the Indonesian economy severely, but it had differential impacts on regional economies”. Against this background, we conducted a bi-dimensional inequality decomposition analysis using provincial GRDP by industrial sector to examine structural changes and inter-provincial income inequality. No previous studies have examined the impacts of the COVID-19 in this manner. Thus, our study contributes to the existing body of literature.

Comment 3

Absence of a literature review section (the authors could have used an excerpt from the current introduction and expanded it with the research results of other authors)

Response

We removed the description of the review of literatures from the introduction and created a new section on literature review, in which we refer to some additional articles on the effects of the COVID-19 on income inequality in Indonesia in footnote 4.

Comment 4

Absence in the description of the methods of justification for the chosen research period (2010-2020?)

Response

We added the following sentence in the method section. “The method we employed enables us to comprehensively assess the effects of changes in both industrial and spatial structure on regional income inequality. Consequently, it offers a robust framework for examining the effects of the COVID-19 on inter-provincial income inequality.”

Comment 5

Too superficial discussion of the results (the authors should confront the obtained results with the results of other authors, as well as refer to the research questions and research hypotheses)

Response

Since our research is descriptive, we do not aim to perform any hypothesis testing. Since no previous studies have examined the impacts of the COVID-19 on both industrial and spatial structure in a comprehensive manner, we are not able to compare our results with the results of other researchers. But, in the conclusion, we added the following sentence after “Many provinces, particularly those relying on tourism experienced a large negative growth, while some poorer provinces in the eastern part of Indonesia escaped from a severe economic downturn.”

“This finding is consistent with the study conducted by Gibson and Olivia (2020), in which richer provinces with lower initial poverty headcount ratios, such as Bali and Riau Islands, tend to exhibit a larger increase in poverty incidence while poorer provinces in the eastern part of Indonesia, such as Papua and North Maluku, tend to experience a smaller increase in poverty incidence.”

Comment 6

Misconstrued summary (the authors should refer to the research gaps, main aim and research hypotheses, then they should present the most important conclusions, indicate what were the limitations of the conducted study and indicate future research directions).

Response

In the conclusion, we added the following paragraph. “While our study provides valuable insights into the initial impacts of the COVID-19 pandemic on inter-provincial income inequality, it is not without limitations. First, the bi-dimensional inequality decomposition method that we employed is descriptive. Thus, in future research, we plan to conduct an econometric analysis using panel data for 33 provinces to assess the effects of the pandemic on provincial economies. Second, due to the unavailability of provincial GDP data by 52 industrial sectors for the period after 2021, we could not analyze the extended effects of the COVID-19 on inter-provincial income inequality in a comparable analytical framework. Thus, we plan to conduct further research to investigate the longer-term effects using provincial GDP data by 52 industrial sectors for the period from 2019 to 2023.”

Comment 7

Article does not meet the editorial requirements of the journal.

Response

We are not qualified to respond to this comment.

Reviewer 3 Report

This paper illustrates the effect of COVID-19 on the Indonesian regional economy. Using provincial GDP by industrial sectors and incorporating a bi-dimensional inequality decomposition analysis, the authors examined how structural changes affect inter-provincial inequality. A few things need to be noticed before publishing the manuscript:

1. Why should we bother with regional inequality in Indonesia? The motivation of the study needs to be clear.

2. Except self-citations, the authors did not cite other papers that worked on inequality in Indonesia. The authors can consider other studies that worked on the same country's inequality. For example, Chongvilaivan and Kim (2016) and Thye et al. (2022).

3. Line diagram is not suitable for cross-sectional data. Figure 2, 5, and 7 can be revised accordingly.

4. Perhaps, the paper was drafted during the COVID pandemic. We have already overcome that situation. Therefore, the authors should address this thing throughout the paper. For instance, the statement “When Indonesia will recover from the pandemic, …..” should be revised.

Reference

Chongvilaivan, A., and Kim, J. (2016). Individual Income Inequality and Its Drivers in Indonesia: A Theil Decomposition Reassessment. Social Indicators Research, 126(1), 79–98. https://doi.org/10.1007/s11205-015-0890-0

Thye, G. L., Law, S. H., and Trinugroho, I. (2022). Human capital development and income inequality in Indonesia: Evidence from a nonlinear autoregressive distributed lag (NARDL) analysis. Cogent Economics & Finance, 10(1). https://doi.org/10.1080/23322039.2022.2129372 

Author Response

Comment 1

Why should we bother with regional inequality in Indonesia? The motivation of the study needs to be clear.

Response

We added the following (underline) in the introduction. “On the other hand, numerous studies have been conducted to analyze regional income inequality in Indonesia using provincial GDP because a large income disparity has persisted between provinces.” We also added the following in footnote 4. “In 2022, Jakarta has the largest per capita GDP at 298 million Rupiah, which is 14 times the smallest in East Nusa Tenggara.”

Comment 2

Except self-citations, the authors did not cite other papers that worked on inequality in Indonesia. The authors can consider other studies that worked on the same country's inequality. For example, Chongvilaivan and Kim (2016) and Thye et al. (2022).

Response

We added Esmara (1975) in the literature survey, as it is the pioneering study on provincial regional income inequality. There have been numerous studies on income (or expenditure) inequality in Indonesia. But our study focused on inequality in per capita GDP between provinces. Thus, it did not include papers on income (or expenditure) inequality using household survey data (National Socio-economic Survey or Family Life Survey (IFLS)). Chongvilaivan and Kim (2016) and Thye et al. (2022) used household survey data to analyze inter-personal income inequality, not inter-provincial income inequality.

Comment 3

Line diagram is not suitable for cross-sectional data. Figure 2, 5, and 7 can be revised accordingly.

Response

In Figure 2, bar diagram is used. In Figures 5 and 7, we eliminated growth rates of per capita GDP because Figures 6 and 8 presented them in scatter plots. Thus, in Figures 5 and 7, bar diagram is only used.

Comment 4

Perhaps, the paper was drafted during the COVID pandemic. We have already overcome that situation. Therefore, the authors should address this thing throughout the paper. For instance, the statement “When Indonesia will recover from the pandemic, …..” should be revised.

Response

Because we wanted to examine immediate impacts of the COVID-19 on provincial economies, we used “The Initial Impacts of the COVID-19 Pandemic …” as the title of our paper. Because we are not able to obtain 52-sector GRDP data after 2021, we could not analyze the effects of the COVID-19 on regional economies in recent years.  In future research, we would like to investigate structural changes and inter-provincial income inequality after Indonesia recovered from the COVID-19.

Round 2

Reviewer 1 Report

The author (s) has incorporated all comments and the article is publishable in the journal. 

Author Response

Thank you very much for your valuable comments.

Reviewer 2 Report

Thank you for the changes made. I make no further comments.

Author Response

(The authors gave the same response as above.)
